# Antibacterial and Antibiofilm Effects of L-Carnitine-Fumarate on Oral Streptococcal Strains *Streptococcus mutans* and *Streptococcus sobrinus*

**DOI:** 10.3390/microorganisms12081613

**Published:** 2024-08-07

**Authors:** Anna Goc, Waldemar Sumera, Matthias Rath, Aleksandra Niedzwiecki

**Affiliations:** Dr. Rath Research Institute, 5941 Optical Ct., San Jose, CA 95138, USA; w.sumera@drrath.com (W.S.); drrath@drrath.org (M.R.)

**Keywords:** L-carnitine, *Streptococcus mutans*, biofilm

## Abstract

*Streptococcus mutans* is a major pathogenic habitant of oral caries. Owing to its physiological and biochemical features, it prevails in the form of plaque biofilm together with another important mutans streptococci species, *Streptococcus sobrinus*. Both species are considered as initiators of cavity lesions, and biofilm is essential to the dental caries process. Compared with the planktonic populations, the biofilm form has higher resistance to environmental conditions and antibiotics. Dental plaques also secure the long-term survival of microorganisms and protection from any stress conditions. To address the need for new antibiofilm agents, we have focused on L-carnitine-fumarate, a fumarate-conjugated quaternary ammonium compound. Using the macro-broth susceptibility testing method, we established its MIC value as 6.0 mg/mL. The MBC value, determined from the broth dilution minimum inhibitory concentration test by sub-culturing it to BHI agar plates, was established as 7.0 mg/mL. Antibiofilm efficacy was tested in 96-well plates coated with saliva using BHI broth supplemented with 1% sucrose as a standard approach. The obtained results allowed us to assess the MIBC as 7.5 mg/mL and the MBBC value as 10.0 mg/mL. The latter concentration also caused approximately 20% eradication of pre-existing biofilm. EPS-rich matrix, forming the core of the biofilm and enabling a confined acidic microenvironment, was also examined and confirmed the effectiveness of 10.0 mg/mL L-carnitine-fumarate concentration in inhibiting EPS formation. Furthermore, the anti-adherent and anti-aciduric impacts of L-carnitine-fumarate were investigated and revealed significant inhibitory effects at sub-MIC concentrations. The influence of L-carnitine-fumarate on the phosphotransferase system was investigated as well. Our results provide a new insight into the antibacterial potential of L-carnitine-fumarate as a valuable compound to be considered for alternative or adjunct anti-caries and antibiofilm preventive approaches.

## 1. Introduction

*Streptococcus mutans* is a Gram-positive, acidogenic, and aciduric coccus isolated in 1924 by J. Clarke from carious lesions. In the 1960s, clinical and animal studies concluded in recognizing *S. mutans* as a key and significant etiologic agent of dental caries. The ordinary habitat of *S. mutans* is biofilm (dental plaque), which is shared with other bacterial species, e.g., *Streptococcus sobrinus* and *Streptococcus sanguinis* [1,2,3]. However, *S. mutans* is considered as a main pathogenic factor in the development of dental caries, which fashions an advantageous niche for other species to thrive, by forming an EPS-rich and low-pH environment. Strains of *S. mutans* have been classified into four serological groups (i.e., c, e, f and k). Nonetheless, ~75% of strains isolated from dental plaque are classified as serotype c, and the serotype c *S. mutans* strain UA159 has been utilized as an experimental bacterial model since its complete genome sequence was published in 2001 [4,5,6,7].

Biofilm is a superb environment for bacterial survival, allowing for their adhesion and cohesion (the aggregation of bacterial cells), being at the same time a reservoir of energy [7,8]. The building blocks of biofilm are glucan (10–20% dry weight), fructan (1–2% dry weight), and proteins (40% dry weight), forming an amorphous three-dimensional structure of exopolysaccharide matrix (EPS) that, in situ, is 80% water [9,10,11,12,13,14]. It is a well-recognized fact that the cariogenic potential of *S. mutans* is attributed to the formation of biofilm on the surface of the tooth, metabolizing a wide variety of carbohydrates into organic acids (acidogenicity) with the ability to survive in a low-pH milieu (aciduricity). The development of biofilm initiates with the layering of the tooth surface with the salivary pellicle formed by salivary components (e.g., proline-rich proteins, amylase, lysozyme, mucin), and bacterial components (e.g., glucosyltransferase, fructosyltransferase, lipoteichoic acid). Bacterial glucosyltransferases (Gtfs) and fructosyltransferases (Ftfs) are the principal sources of extracellular polymeric substances (EPSs) as a product of sucrose and starch hydrolysates [15,16,17]. Together with surface-associated glucan-binding proteins (Gbps) encoded by *S. mutans*, Gtfs and glucans aid sucrose-dependent biofilm formation and promote the accumulation of microbial cells. In addition, *S. mutans* expresses multiple high-affinity surface adhesins that allow its colonization in the absence of sucrose. Another functional constituent of biofilm is eDNA (environmental DNA), which forms nanofibers that support the coherence of bacterial cells and their adherence to the substratum, thereby contributing to the structural integrity and stability of a biofilm [18].

L-carnitine (LC), discovered in 1905 by Gulewitsch and Krimberg, and by Kutscher, is an amino acid derivative naturally occurring in many different types of food, but also endogenously synthesized in all kinds of eucaryotic genres. In humans, it is produced mainly from L-lysine in the brain, liver, and kidneys, when not provided in the diet, and ~95% is stored in the heart and skeletal muscle [19,20,21]. LC is a substrate for its conversion into acetyl-L-carnitine and propionyl-L-carnitine, as well as an essential co-factor for the enzymes involved in the transportation of the long-chain fatty acids into the mitochondria, where they are subsequently used for energy production by beta-oxidation [20,21,22]. Half a century after the discovery of carnitine, it was established that various aerobic or anaerobic Gram-positive and Gram-negative bacteria may utilize carnitine in different cellular processes, including those involved in helping to survive environmental insults, for energy production, or serving as an osmolyte [20,23,24,25,26]. L-carnitine is an amphiphilic compound, which, as with any such agent, can have effective antimicrobial properties. Olgun et al. reported a moderated antimicrobial effect against a wide range of Gram-positive and Gram-negative bacteria and fungi, while ammonium L-carnitine esters with long alkyl chains and synthetic acylcarnitine analogs were also found to have a mild antibacterial effect [27,28,29]. Owing to the slower development of antibiotics, at the same time as bacterial resistance rapidly intensifies, more microbiological insight about natural antibacterial compounds can only be beneficial. Particularly significant may be information about the antibiofilm properties of compounds, since dental plaque is acknowledged as the main cause of oral pathologies including caries and periodontitis. Thus far, there are no published data suggesting the antimicrobial effect of L-carnitine itself. Reports about the role and significance of L-carnitine in *Streptococcus* spp., particularly in oral streptococci, are scarce as well. To shed more light, particularly on the topic of antimicrobial properties of L-carnitine, we investigated the effect of its two forms (i.e., L-carnitine-fumarate and L-carnitine HCl) against oral *S. mutans* and *S. sobrinus,* focusing on various aspects relevant to their virulence and prevalence in dental plaques.

## 2. Materials and Methods

### 2.1. Compounds, Bacterial Strains, and Culture Conditions

*S. mutans* strain UA159 and *S. sobrinus* strain SL1 were obtained from the American Type Culture Collection (ATCC, Manassas, VA, USA) and maintained on brain heart infusion (BHI) (Difco Laboratories, Detroit, MI, USA) agar plates, and both were recurrently cultured in BHI medium at 37 °C in a 5% CO_2_ incubator. L-carnitine-fumarate (LC) and L-carnitine HCl were obtained from Sigma (St. Louis, MO, USA), and dissolved in 1 × PBS (phosphate-buffered saline) at a 100 mg/mL concentration that served as a stock solution for further testing.

### 2.2. Minimum Inhibitory Concentration (MIC) and Minimum Bactericidal Concentration (MBC) Studies

L-carnitine-fumarate and L-carnitine HCl were examined for minimum inhibitory concentration (MIC) and minimal bactericidal concentration (MBC) against *S. mutans* UA159 and *S. sobrinus* SL1 strains, respectively, as well as their dual-species culture, according to the Clinical Laboratory Standards Institute (CLSI) and/or M100-S23 guidelines for microtiter broth dilution testing [30,31]. These test bacteria were cultured overnight in BHI broth (BD, Sparks, MD, USA) at 37 °C for further experiments. Overnight bacterial cultures were standardized by OD_600_ to 5 × 10^5^ CFU/mL. To evaluate the inhibitory effect of LC on a dual-species culture of test oral streptococci, 2.5 × 10^5^ CFU of one bacterial cell suspension was mixed with 2.5 × 10^5^ CFU of another bacterial cell suspension in BHI broth. Serial dilutions of LC at test a range of 10–1.25 mg/mL were prepared and applied. Next, tubes were incubated at 37 °C for 24 h under static conditions. Bacterial growth was determined by measuring the turbidity (OD_600_) of tubes, and a MIC_90_ was determined. The MIC_90_ value of planktonic cells refers to at least a 90% inhibitory effect compared to a control, established as the difference between the absorbance of the experimental group and the control group in a percentage compared to the control. At the same time, 100 µL of each culture was plated on BHI agar plates for the next 24 h at 37 °C under static conditions to establish minimal bactericidal concentrations. Colonies on each plate were counted using a colony counter, and MBC_90_ values were assigned based on the concentration where at least 90% biocidal effect was observed. The control included the bacterial culture supplemented with the vehicle (10% *v*/*v* 1 × PBS) only. All LC concentrations were tested in triplicate, and tests were repeated three times on different days.

### 2.3. Kinetic Study of LC on Planktonic Form

Time-dependent experiments were performed starting with separate bacterial cultures of *S. mutans* UA159 and *S. sobrinus* SL1 strains, as well as their dual-species culture, first grown overnight in BHI medium, which were then diluted to 1:100 in fresh BHI. Different concentrations of LC (as specified in the figure legends) were prepared and applied. Cell growth was monitored by measuring OD_600_ at 1 h, 3 h, 6 h, 12 h, and finally 24 h intervals using a Cell Density Meter Ultrospec 10 (Amersham Bioscience, Piscataway, NJ, USA). Parallelly, at the same time points, 100 µL of each culture was plated on BHI agar plates and incubated for the next 24 h at 37 °C under static conditions. Colonies on each plate were counted, and obtained values were calculated. The control included the vehicle (10% 1 × PBS) only. All concentrations were tested in triplicate, and the tests were repeated three times on different days.

### 2.4. Minimum Biofilm Inhibitory Concentration (MBIC) and Minimum Biofilm Biocidal Concentration (MBBC) Studies

L-carnitine-fumarate and L-carnitine HCl were examined for minimum biofilm inhibitory concentrations (MBICs) and minimum biofilm bactericidal concentrations (MBBCs) against *S. mutans* UA159 and *S. sobrinus* SL1 strains, respectively, as well as their dual-species culture, according to the methodology previously described [32]. These test bacteria were cultured overnight in BHI broth (BD, Sparks, MD, USA) at 37 °C for further experiments. Overnight bacterial cultures were standardized by OD_600_ to 5 × 10^5^ CFU/mL. To evaluate the inhibitory effect of LC on a dual-species culture of test oral streptococci, 2.5 × 10^5^ CFU of one bacterial cell suspension was mixed with 2.5 × 10^5^ CFU of another bacterial cell suspension in BHI broth. Serial dilutions of LC at a test range of 10–2.5 mg/mL were prepared and applied. Next, 96-well flat-bottom plates coated with human saliva (Innovative Research Inc., Novi, MI, USA) containing bacteria in BHI broth (100 µL), supplemented with 1% sucrose together with different concentrations of LC, were incubated at 37 °C for 24 h under static conditions. Next, the medium was removed, and the wells were washed twice with sterile 1 × PBS. The plates were then subjected to an alamarBlue viability assay and crystal violet staining in order to establish MBICs. To find MBBCs, bacterial cultures were first placed into 96-well plates coated with human saliva (Innovative Research Inc., Novi, MI, USA). After 24 h incubation at 37 °C in a 5% CO_2_ incubator, serial dilutions of LC at a test range of 10–2.5 mg/mL were prepared and then applied on already grown/pre-existing biofilm washed with 1 × PBS. Next, plates containing bacterial biofilms in BHI broth supplemented with 1% sucrose and different concentrations of LC were incubated at 37 °C for an additional 24 h under static conditions. Next, the liquid medium was removed, and the wells were rinsed twice with sterile 1 × PBS. The plates were then subjected to an alamarBlue viability assay and crystal violet staining. The control included the vehicle (10% 1 × PBS) only. The control included the vehicle (10% 1 × PBS) only. All concentrations were tested in quadruplicates, and the tests were repeated three times on different days.

### 2.5. Kinetic Study of LC on Biofilm Formation

Time-dependent experiments were performed starting with the bacterial cultures of *S. mutans* UA159 and *S. sobrinus* SL1 strains, separately, as well as their dual-species culture first grown overnight in BHI medium, which were then diluted to 1:100 in fresh BHI. Different concentrations of LC (as specified in the figure legends) were prepared and applied. Biofilm growth was monitored by measuring RFU_535/595_ (alamarBlue assay) and OD_595_ (crystal violet assay) at 3 h, 6 h, 12 h, and finally 24 h intervals, using a spectrofluorometer (Molecular Device Inc., San Jose, CA, USA). Parallelly, at the same time points, 100 µL of each biofilm culture was gently detached and plated on BHI agar plates and incubated for the next 24 h at 37 °C under static conditions. Colonies on each plate were then counted, and obtained values were calculated. The control included the vehicle (10% 1 × PBS) only. All concentrations were tested in triplicate, and the tests were repeated three times on different days.

### 2.6. Biofilm Viability Assays

The viability of biofilm was quantified as previously described [32]. Briefly, flat-bottom 96-well plates coated with human saliva (two or more donors, pooled) and containing either 100 µL of bacterial culture or pre-existing biofilm were treated with designated concentrations of LC for a designated period of time and supplemented with 20 µL of ready-to-use alamarBlue reagent (Thermo Fisher, Waltham, MA, USA). Next, plates were incubated at 37 °C for an additional 15 min, and the fluorescence was measured at RFU_535/595_ by a spectrofluorometer (Molecular Device Inc., San Jose, CA, USA). The control included the vehicle (10% 1 × PBS) only. The alamarBlue assay was performed in quadruplicates and repeated three times on different days.

### 2.7. Crystal Violet Staining

Biofilm formation and eradication were quantified as previously described [32]. Briefly, flat-bottom 96-well plates coated with human saliva and containing either 100 µL of bacterial culture or pre-existing biofilm were treated with designated concentrations of LC and incubated for a designated period. Next, the medium was removed, and plates were washed twice using sterile 1 × PBS. Then, biofilms in the wells were fixed with 4% (*w*/*v*) paraformaldehyde for 15 min followed by the addition of 0.1% (*w*/*v*) crystal violet solution to each well for another 5 min. After removing the staining solution, the plate was washed three times with sterile water. A 100% (*v*/*v*) methanol was added to dissolve the dye, and the absorbance at OD_575_ was measured with a spectrometer (Molecular Device Inc., San Jose, CA, USA). The control included the vehicle (10% 1 × PBS) only. Crystal violet staining was performed in quadruplicates, each concentration was tested, and the tests were repeated three times on different days. Additionally, an experiment was executed, in which, instead of human-saliva-coated plates, human-saliva-coated artificial teeth were used (Azdent, Baku, Azerbaijan) and were subjected to the same treatment and staining procedures as 96-well plates coated with human saliva.

### 2.8. The Phosphoenolpyruvate (PEP): Carbohydrate Phosphotransferase System (PTS) Assay

*S. mutans* UA159 strain was first cultured overnight in BHI medium, then diluted 1:50 in fresh BHI supplemented with 1% sucrose and different concentrations of LC at a test range of 10–2.5 mg/mL. Next, cultures were incubated at 37 °C for 6 h and 12 h and harvested by centrifugation. The amount of intracellular phosphoenolpyruvate was assessed with a Phosphoenolpyruvate Assay Kit (Cell BioLabs Inc., San Diego, CA, USA), carried out according to the manufacturer’s protocol. Each experiment was performed in quadruplicates and repeated three times on different days.

### 2.9. Qualitative Determination of EPS

For imaging the exopolysaccharide (EPS) presence on biofilms of *S. mutans* UA159, pre-existing biofilms grown on 8-well Nunc™ Lab-Tek™ Chambered Coverglass (Thermo Fisher, Waltham, MA, USA) were treated for 24 h with different concentrations of LC (10–2.5 mg/mL), stained with 1.0 µM Alexa Fluor 633-labeled concanavalin A, and immersed in ultra-pure water for up to 1 h, as previously reported [33]. The fluorescent dye was excited with a HeNe laser (633 nm), and images were captured with a fluorescence microscope Zeiss Axio Observer 3 (Carl Zeiss Microscopy LLC, White Plains, NY, USA). Estimations of polysaccharides and protein analysis were performed in biofilms of *S. mutans* UA159 grown in 24-well plates, previously coated with human saliva and treated with different concentrations of LC (10–2.5 mg/mL) for 24 h. Next, 1 × PBS was added to the biofilms, which were then gathered using sonication/vortexing, and the amount of carbohydrates was determined by the anthrone method (glucose was used as the standard [34]), while the total amount of protein was assessed with a BCA Protein Assay Kit according to the manufacturer’s protocol (bovine serum albumin was used as the standard (Thermo Fisher, Waltham, MA, USA). To assess the biomasses of biofilms, an aliquot of the sonicated biofilm suspensions was added to previously weighed micro-centrifuge tubes, air-dried, and weighed to determine the difference between the final and primary weight of the micro-centrifuge tubes [32]. The assessment of the amounts of extracellular (soluble and insoluble) polysaccharides was performed according to published methodology [33,34,35,36]. Briefly, an aliquot of 0.3 mL of the sonicated biofilm suspensions was centrifuged at 10,000 g for 10 min at 4 °C. The supernatants were collected, and the biofilm pellets were washed two times with sterile water followed by centrifugation at 10,000 g for 10 min for separation of water-soluble (supernatant) from water-insoluble polysaccharides (pellet). Three parts of cold methanol were added to collected supernatants, and precipitates were centrifuged, air-dried, and resuspended in water. Next, the amount of water-soluble polysaccharides was assessed by the anthrone method, whereas the pellets were dissolved in 1 M NaOH (1 mg of pellet/0.3 mL of 1 M NaOH), and the water-insoluble polysaccharides were then quantified in the same way as water-soluble polysaccharides. The absorbance was measured at 625 nm, and the concentrations of water-soluble and water-insoluble polysaccharides were calculated using standard curves. Each experiment was performed in quadruplicates and repeated three times on different days.

### 2.10. Aciduricity Assay

The impact of LC on the acid tolerance of *S. mutans* UA159 was examined by measurement of the viability of bacteria after 2 h of exposure at pH = 5.0 as previously described [36,37]. Briefly, *S. mutans* UA159 was grown in TYEG (tryptone–yeast extract supplemented with 20 mM glucose) medium until the mid-logarithmic phase, equal to OD_600_ = 0.5. The cells were collected by centrifugation and resuspended to OD_600_ = 0.2 in TYEG medium buffered with 20 mM phosphate–citrate buffer (pH = 5.0) containing different concentrations of LC and incubated at 37 °C for 2 h. The control tubes contained equivalent amounts of 1 × PBS only. Samples were removed before and after incubation at pH = 5.0 for viable counts of bacterial colonies on BHI agar plates. Each experiment was performed in triplicates and repeated three times on different days.

### 2.11. Adherence Assay

Sucrose-dependent and sucrose-independent glass surface adherence assays were performed according to a published report [38,39]. Briefly, overnight culture of *S. mutans* UA159 was diluted to 1:100 in fresh BHI broth (without sucrose or containing 5.0% sucrose). Next, 5 mL of each culture was treated with different concentrations of LC ranging from 10 to 2.5 mg/mL. All samples were then allowed to grow for 24 h at 37 °C at an angle of 30 degrees in glass tubes. The control tubes contained BHI (with or without sucrose) and equivalent amounts of 1 × PBS. After incubation, non-adherent cells were removed, and the attached bacterial cells were detached with 0.5 M KOH. Adherence was quantified by recording changes in OD_600_ as follows: percentage adherence = [OD_600_ of adhered cells/OD_600_ of adhered cells + OD_600_ of supernatant cells] × 100. Each experiment was performed in triplicates and repeated three times on different days. Parallelly, adherence on plastic plates not coated with human saliva was assessed as previously described [33,40]. Briefly, bacterial cultures diluted to 1:100 in fresh BHI broth (containing 5.0% sucrose) were added to 96-well plates and treated with different concentrations of LC ranging from 10 to 2.5 mg/mL. At different time points (i.e., 6 h, 12 h, and 24 h), unattached bacterial cells were removed, and the remaining attached biofilms were quantified with the crystal violet staining method. Each experiment was performed in triplicates and repeated three times on different days.

### 2.12. Statistical Analysis

All data are presented as means ± SD. All experiments were performed at least three times, each at least in triplicates. Student’s two-tailed *t*-test was used to determine statistically significant differences set at the 0.05 level. Statistical analysis was performed using GraphPad software.

## 3. Results

### 3.1. Bacteriostatic and Bactericidal Effect of L-Carnitine Forms on the Planktonic Form of Oral Streptococci

To assess the effect of L-carnitine-fumarate and L-carnitine HCl on *S. mutans* and *S. sobrinus*, we first performed concentration-dependent treatment of their respective planktonic forms and their together-mixed cultures. Using the macro-broth susceptibility testing method, we were able to establish MIC values of 6.0 mg/mL (for *S. mutans* and for *S. mutans* + *S. sobrinus*) and 5.5 mg/mL (for *S. sobrinus*), as presented in Figure 1A. A supportive experiment, where L-carnitine HCl was applied (as an example of another form of L-carnitine), following the same treatment regimen, was performed and revealed no difference in susceptibility (Table 1). The kinetic evaluation of sub-MIC (i.e., 3.0 mg/mL) and MIC (i.e., 6.0 mg/mL) values showed a ~99% decrease in OD_600_ in the growth of both respective planktonic cultures and their together-mixed cultures after 1 h of incubation with 6.0 mg/mL L-carnitine-fumarate. The same experiment revealed a ~50% decrease in OD_600_ in the growth of both respective planktonic cultures and their together-mixed cultures after 6 h of incubation with 3.0 mg/mL L-carnitine-fumarate (Figure 1B). Additionally, the biocidal effect was tested from a broth dilution minimum inhibitory concentration experiment, by sub-culturing bacterial planktonic cultures treated with different concentrations of L-carnitine-fumarate to BHI agar plates, which allowed us to establish MBC values as 7.0 mg/mL (for *S. mutans* and for *S. mutans* + *S. sobrinus*) and 6.5 mg/mL (for *S. sobrinus*), as presented in Table 1. A supportive experiment, where L-carnitine HCl was applied, following the same treatment regimen, was performed and revealed no difference in susceptibility (Table 1). The time-dependent experiment further showed a ≥2.0-log CFU mg/mL decrease (~99–99.9%% bactericidal effect) of both respective planktonic cultures and their together-mixed cultures after 1 h of incubation with 7.0 mg/mL L-carnitine-fumarate (~2.0-log CFU mg/mL for *S. mutans* and for *S. mutans* + *S. sobrinus* and a ~2.9-log CFU mg/mL decrease for *S. sobrinus*). In the same experiment, we observed a ~5.0-log CFU mg/mL decrease (~99.999% bacteriostatic effect) of both respective planktonic cultures and their together-mixed cultures after 3 h of incubation with 7.0 mg/mL L-carnitine-fumarate, and a ~1.0-log_10_ CFU mg/mL decrease (~90% bacteriostatic effect) after 6 h incubation with 5.0 mg/mL L-carnitine-fumarate, which did not increase with up to 24 h incubation (Figure 1C).

### 3.2. Bacteriostatic and Bactericidal Effect of L-Carnitine Forms on the Biofilm Form of Oral Streptococci

In addition to testing the bacteriostatic and bactericidal effects of L-carnitine-fumarate and L-carnitine HCl on the biofilm form of *S. mutans* and *S. sobrinus*, we performed concentration-dependent treatment of their respective biofilm forms and their together-mixed biofilm cultures. Using bacterial cultures in BHI broth supplemented with 1% sucrose, we applied them to 96-well plates coated with human saliva and subjected them to alamarBlue and crystal violet assays. We were able to establish MBIC values as 7.5 mg/mL (for *S. mutans* and for *S. mutans* + *S. sobrinus*) and 6.5 mg/mL (for *S. sobrinus*), as presented in Figure 2A,B. A parallel experiment performed on artificial teeth revealed the same pattern. A confirmatory experiment, where L-carnitine HCl was applied instead of L-carnitine-fumarate, following the same treatment regimen, was performed and revealed no difference in susceptibility (Table 1). The kinetic evaluation of sub-MIC (i.e., 5.0 mg/mL) and MBIC (i.e., 7.5 mg/mL) values showed ~25% reduced metabolic activity of both respective biofilm cultures and their together-mixed cultures after 3 h of incubation with 5.0 mg/mL L-carnitine-fumarate. A similar experiment revealed ~90% reduced metabolic activity of both respective planktonic cultures and their together-mixed cultures after 3 h of incubation with 7.5 mg/mL L-carnitine-fumarate (Figure 2C). In addition, the experiment in which biofilm formation was monitored with CV staining revealed a ~45% inhibition in the biofilm formation of *S. mutans*, *S. sobrinus*, and *S. mutans* + *S. sobrinus* after 3 h treatment with 5.0 mg/mL L-carnitine-fumarate, and ~95% inhibition in biofilm formation of *S. mutans*, *S. sobrinus*, and *S. mutans* + *S. sobrinus* after 3 h treatment with 7.5 mg/mL L-carnitine-fumarate (Figure 2D). A parallel experiment performed on artificial teeth revealed the same pattern.

In the experiments where pre-existing biofilm growth in BHI broth supplemented with 1% sucrose in 96-well plates coated with human saliva was a subject of study, we were able to establish the MBBC value as 7.5 mg/mL (for *S. mutans*, *S. sobrinus*, *S. mutans+S. sobrinus*), as presented in Figure 3A. A corroborative experiment, where L-carnitine HCl was applied instead of L-carnitine-fumarate, following the same treatment regimen, was performed and revealed no difference in susceptibility (Table 1). L-carnitine-fumarate tested in a time-dependent pattern of pre-existing biofilms of *Streptococcus* spp. was shown to be biocidal at 7.5 mg/mL after 6 h, reducing the viability of pre-existing biofilm by 25% after 3 h (Figure 3B). A parallel experiment, in which biofilms were first treated with 7.5 mg/mL L-carnitine-fumarate and then left for additional 24 h to allow for recovery, confirmed that this concentration is biocidal at the rate of ≥99.0% after 6 h and with approximately 25% efficacy after 1 h, as measured by the alamarBlue method (Figure 3C). Nonetheless, decreased viability of pre-existing biofilm of *S. mutans* and *S. sobrinus,* as well as their mixed biofilm cultures, was observed at sub-MIC concentrations of L-carnitine-fumarate (i.e., 5.0 mg/mL).

Furthermore, this biocidal concentration seems to mildly augment the detachment of both pre-existing *Streptococcus* spp. biofilms and their mixed biofilm, being slightly more profound at a higher (i.e., 10 mg/mL) L-carnitine-fumarate concentration (Figure 4A,B). Interestingly, we noticed that the MBBCs of L-carnitine-fumarate revealed a greater detachment effect of pre-existing biofilms after 24 h treatment and further 24 h biofilm culturing, this time in the absence of the test compound (Figure 4B). Pre-existing biofilms treated with L-carnitine-fumarate for 24 h were shown to be “thinner” in response to higher L-carnitine-concentrations, which resulted in reduced biofilm biomass and lesser amounts of total protein as well as lesser amounts of water-soluble and water-insoluble polysaccharides (Table 2). The effect was rather mild at sub-MIC concentrations of L-carnitine-fumarate (i.e., 5.0 mg/mL), although statistically significant at MBBCs (Table 2). The qualitative and quantitative analysis of EPS staining showed that, compared with the control, there was a significant reduction in the presence of EPS in 10 mg/mL L-carnitine-fumarate-treated biofilms of *Streptococcus* spp. and their mixed biofilm (Figure 4C and Table 2).

### 3.3. Effect of L-Carnitine-Fumarate on Biochemical and Physiological Processes of S. mutans UA159 Strain

The effect of L-carnitine-fumarate on the viability and detachment of respective biofilms of *S. mutans* and *S. sobrinus* or their mixed biofilm cultures found corroboration in their adherence to glass and plastic surfaces (Figure 5). Treatment with L-carnitine-fumarate resulted in reduced sucrose-independent and sucrose-dependent adherence of bacteria in a dose-dependent manner. The sub-MIC concentration of L-carnitine-fumarate (i.e., 5.0 mg/mL) 100% inhibited sucrose-independent adherence of *S. mutans* UA159 to a glass surface, whereas the same concentration 90% inhibited sucrose-dependent adherence of *S. mutans* UA159 to a glass surface. Adherence to glass in the presence of 5% sucrose was distinctly more evident and completely inhibited by the MBBC (i.e., 7.5 mg/mL). The adherence was also checked at 6, 12, and 24 h on plastic plates not coated with human saliva to check whether it could adversely affect *S. mutans* UA159 biofilm formation at different phases of its biofilm growth: 6 h (adherent phase), 12 h (active accumulated phase), and 24 h (plateau accumulated phase) [40]. This experiment revealed that the effect of L-carnitine-fumarate is indeed concentration-dependent, but not so much biofilm phase growth-dependent. The percentage of adherent bacteria gradually accumulated up to 24 h in the control and began to decline sharply from >2.5 mg/mL, with the concentration of L-carnitine-fumarate being statistically significant (51.8% of inhibition after 12 h and 42.3% of inhibition after 24 h) at sub-MIC concentration (i.e., 5.0 mg/mL). Inhibition of biofilm formation by L-carnitine-fumarate during different phases of biofilm growth was rather constant and completely reduced at MBBC (i.e., 7.5 mg/mL).

Moreover, as shown in Figure 6A, we observed a dose-dependent decline in the survival of *S. mutans* UA159 cells at pH = 5.0, starting from sub-MIC concentrations (i.e., 2.5 mg/mL). At the same time, exposure of its planktonic form at the mid-logarithmic and the late logarithmic/early stationary phase to L-carnitine-fumarate caused mild inhibitory effects on the PEP/PTS of *S. mutans* UA159. As presented in Figure 6B, the amount of phosphoenolpyruvate began to drop significantly from the 2.5 mg/mL concentration of L-carnitine-fumarate that reduced the phosphoenolpyruvate by ~20–28%, and by ~45–60% at 5.0 mg/mL of L-carnitine.

## 4. Discussion

L-carnitine (3-hydroxy-4-N-trimethylammoniobutanoate, β-hydroxy-γ-N-trimethylaminobutyric acid) has driven the attention of research and medical communities since the discovery of its benefits for human health. From the time of these discoveries, our knowledge about carnitine’s function has increased, and further expanded to the field of microbiology, gaining more information about carnitine’s role in bacterial metabolism. Meanwhile, interest in the therapeutic use of carnitine is growing, partially due to the finding of the significance and consequences of its deficit [41,42,43]. Safety concerns of carnitine have been investigated as well [44]. According to the observed safe level (OSL), consumption of <2000 mg/day of LC is not considered to be a risky approach. Intakes of LC >2000 mg/day faced confidence issues with the estimation of long-term safety. Nonetheless, thus far, there is no evidence that high or even very high doses of LC are deemed to be toxic or genotoxic [45]. The pharmacokinetics of LC in healthy volunteers after a single 2.0 g L-carnitine oral administration was studied as well. The maximum plasma concentration (Cmax) of L-carnitine was 84.7 ± 25.2 µmol/L. The elimination half-life of L-carnitine was established as 60.3 ± 15.0 h and the time required to reach the Cmax (i.e., Tmax) as 3.4 ± 0.46 h. [46]. However, another aspect of oral ingestion of L-carnitine needs also to be taken into consideration. Namely, long-term consumption of high doses of L-carnitine and its influence on human microbiota. The results published by Wu et al. imply that intake of high concentrations of L-carnitine indeed changes gut microbiota composition and at the same time causes the enhancement of TMAO (Trimethylamine N-oxide) production [47]. TMAO is an end-product formed from dietary L-carnitine, through the intermediates γBB (γ–Butyrobetaine) and TMA (Trimethylamine), which in humans is involved in atherosclerosis- and thrombosis-related disorders [19,48,49].

Here, we investigated the antibacterial effect of L-carnitine-fumarate against oral streptococcal strains *S. mutans* UA159 and *S. sobrinus* SL1 as well as their mixed-species cultures. We also included supportive results obtained from exploring additional selected mechanisms by which L-carnitine-fumarate impacts the cariogenic processes. Our data showed that applications of 6.5–7.0 mg/mL L-carnitine-fumarate or L-carnitine HCl have a bacteriostatic effect on planktonic forms of *S. mutans* and *S. sobrinus* and their mixed cultures, while 7.0 mg/mL expresses a bactericidal effect. Moreover, these two forms of L-carnitine at 7.5 mg/mL concentrations revealed an inhibitory effect on biofilm growth of the test streptococcal strains as well as a biocidal effect on their already pre-existing and established biofilm forms. Furthermore, with 10 mg/mL of either L-carnitine-fumarate or L-carnitine HCl, we observed mild disruption of pre-existing biofilms, which found its confirmation in affected biofilm biomass and EPS production. While detecting EPS with Alexa Fluor 633-conjugated concanavalin A (i.e., a carbohydrate-binding protein, excitation/emission 632/647 nm), selectively binding α-mannopyranosyl and α-glucopyranosyl residues deposited in the EPS matrix, we noticed impaired accumulation of proteins and carbohydrates (both soluble polysaccharides and insoluble polysaccharides) [50]. It is a well-known fact that biofilm of *S. mutans* is abundant in soluble and insoluble glucans, which provide a framework and structural integrity to the biofilm. At the same time, we noticed that the *S. sobrinus* strain seems to be more susceptible than *S. mutans*. Considering these data together with our experiments on bacterial cells’ adherence to either glass or plastic surfaces, we could establish that L-carnitine-fumarate can interrupt this process regardless of the type of surface (glass vs. plastic), the presence of sucrose (a known compound facilitating biofilm build-up), or the phase of biofilm growth (early, middle, or late stage). It needs to be mentioned, however, that further study utilizing more sophisticated and detrimental methods, such as atomic force microscopy or micromanipulation, could be used and provide more detailed information on this particular topic. In conjunction with the other two experiments focused on aciduricity (i.e., viability of bacterial cells in pH = 5.0) and amount of PEP (the main source of energy for the phosphotransferase system), we also observed declining survival rates upon treatment with L-carnitine-fumarate at pH = 5.0 similar to declining PEP production levels that greatly decreased at biocidal concentrations of L-carnitine-fumarate, compared with a vehicle-treated control. We noticed, however, that regardless of the performed experiment, a decreasing pattern was observed from sub-MIC concentrations (i.e., 2.5 mg/mL).

In conclusion, we have shown that L-carnitine-fumarate and L-carnitine HCl do demonstrate bacteriostatic or bactericidal antibacterial activity against oral streptococcal strains, although high concentrations of these compounds are required.

## Figures and Tables

**Figure 1 microorganisms-12-01613-f001:**
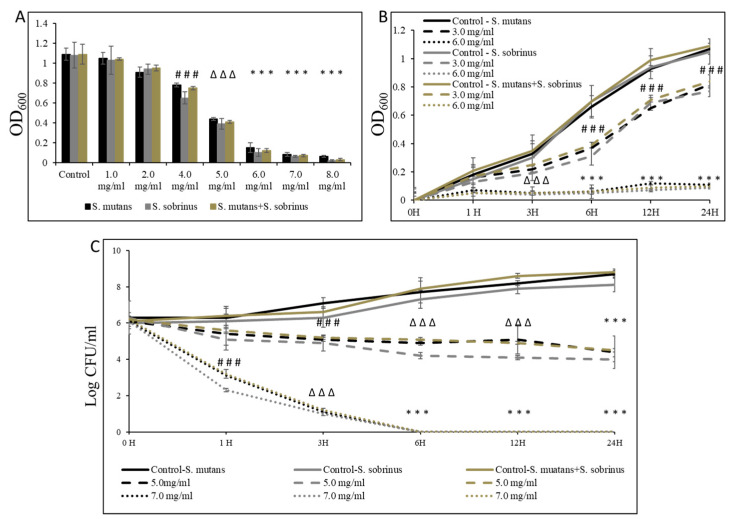
Bacteriostatic and bactericidal efficacy of L-carnitine-fumarate on the planktonic form of *Streptococcus mutans* UA159 and *Streptococcus sobrinus* SL1. Dose-dependent effect of different concentrations of LC evaluated by macro-dilution method at 24 h (**A**). Time-dependent effect of sub-MIC and MIC values of LC assessed by macro-dilution method up to 24 h (**B**). Time-dependent effect of sub-MBC and MBC values of LC assessed by macro-dilution method up to 24 h (**C**). MBC values were determined from broth macro-dilution minimum inhibitory concentration test by sub-culturing it on BHI agar plates that did not contain LC. Control—10% 1 × PBS; LC—L-carnitine-fumarate; # *p* < 0.05, ∆ *p* ≤ 0.01, * *p* < 0.001 compared to control.

**Figure 2 microorganisms-12-01613-f002:**
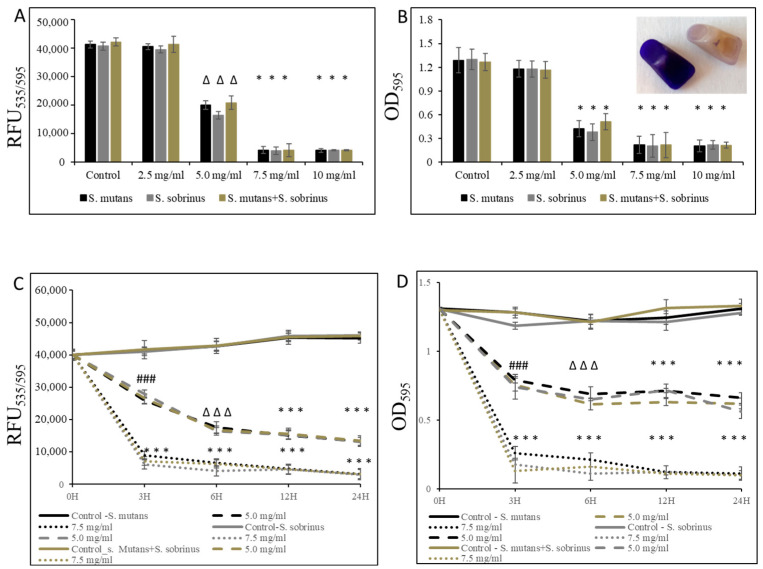
Effect of L-carnitine-fumarate on the biofilm growth of *Streptococcus mutans* UA159 and *Streptococcus sobrinus* SL1. Dose-dependent effect of different concentrations of LC evaluated at 24 h. MBIC values were determined on 96-well plates coated with human saliva after 24 h incubation with LC and assessed by alamarBlue (**A**) and crystal violet (**B**) staining methods. Insert: representative images of biofilms of *S. mutans* formed on artificial teeth coated with human saliva after 24 h incubation period with 7.5 mg/mL LC. Time-dependent effect of sub-MBIC and MBIC values of LC evaluated up to 24 h. MBIC values were determined on 96-well plates coated with human saliva after 24 h incubation with LC and assessed by alamarBlue (**C**) and crystal violet (**D**) staining methods. Control—10% 1 × PBS; LC—L-carnitine-fumarate; # *p* < 0.05, ∆ *p* ≤ 0.01, * *p* < 0.001 compared to control.

**Figure 3 microorganisms-12-01613-f003:**
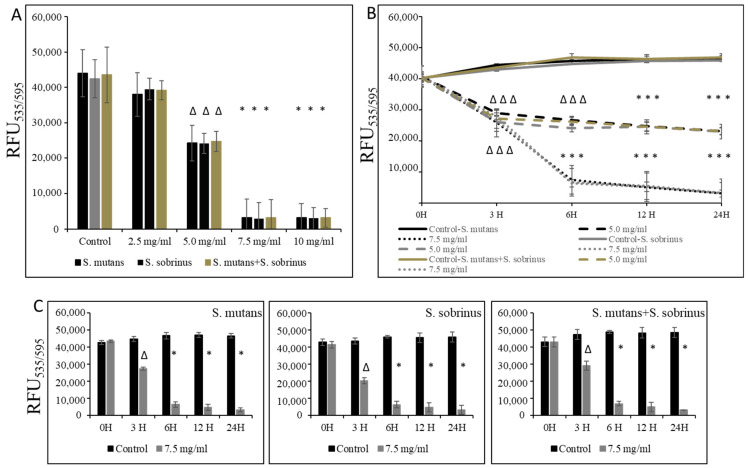
Biocidal effect of L-carnitine-fumarate against the pre-existing biofilm of *Streptococcus mutans* UA159 and *Streptococcus sobrinus* SL1. Dose-dependent effect of different concentrations of LC evaluated at 24 h. MBBC values were determined on 96-well plates coated with human saliva after 24 h incubation with LC and assessed by alamarBlue staining methods (**A**). Time-dependent effect of sub-MBBC and MBBC values of LC evaluated up to 24 h. MBBC values were determined on 96-well plates coated with human saliva after 24 h incubation with LC and assessed by alamarBlue (**B**). Time-dependent effect of MBBC value of LC evaluated after 24 h incubation period in BHI broth supplemented with 1% sucrose only (**C**). Control—10% 1 × PBS; LC—L-carnitine-fumarate;—∆ *p* ≤ 0.01, * *p* < 0.001 compared to control.

**Figure 4 microorganisms-12-01613-f004:**
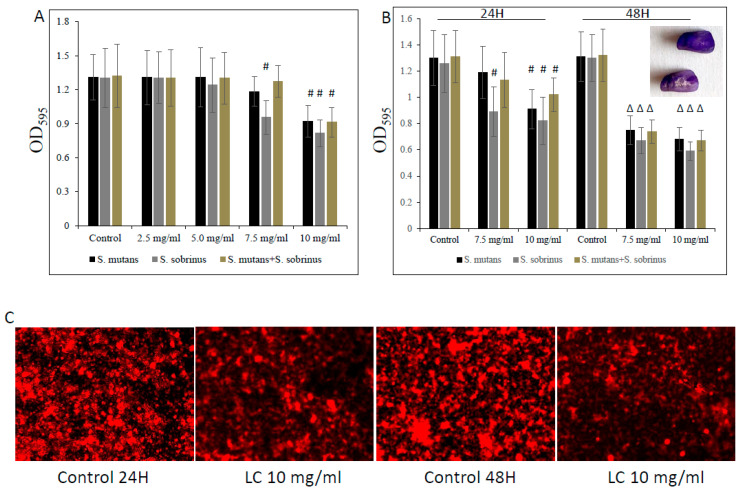
Effect of L-carnitine-fumarate on the detachment of the pre-existing biofilm of *Streptococcus mutans* UA159 and *Streptococcus sobrinus* SL1. Dose-dependent effect of different concentrations of LC evaluated at 24 h. MBEC values were determined on 96-well plates coated with human saliva after 24 h incubation with LC and assessed by crystal violet staining methods (**A**). Effect of 7.5 mg/mL and 10 mg/mL concentrations of LC evaluated at 24 h and at 48 h assessed by crystal violet staining methods (**B**). Insert: representative images of biofilms of *S. mutans* on artificial teeth coated with human saliva after 24 h incubation period with 10 mg/mL LC. Representative images of EPS presence on pre-existing biofilm of *S. mutans* after 24 h incubation period with 10.0 mg/mL LC stained with Alexa Fluor 633-conjugated concanavalin A (**C**). Control—10% 1 × PBS; LC—L-carnitine-fumarate; # *p* < 0.05, ∆ *p* ≤ 0.01, compared to control.

**Figure 5 microorganisms-12-01613-f005:**
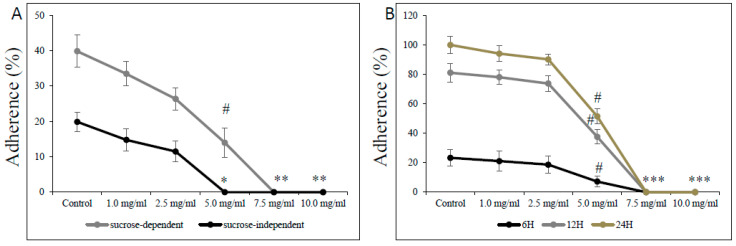
Effect of L-carnitine-fumarate on the adherence of *Streptococcus mutans* UA159. Glass adherence in the absence (sucrose-independent) and presence of 5% sucrose (sucrose-dependent) was evaluated by measuring changes in OD_600_ nm after adding 0.5 M NaOH as described in the Section 2 (**A**). Plastic adherence in the presence of 5% sucrose (sucrose-dependent) was assessed at different time points on 96-well plates not coated with human saliva, by measuring changes in OD_595_ nm (**B**). Control—10% 1 × PBS; LC—L-carnitine-fumarate; # *p* < 0.05, * *p* < 0.001 compared with control.

**Figure 6 microorganisms-12-01613-f006:**
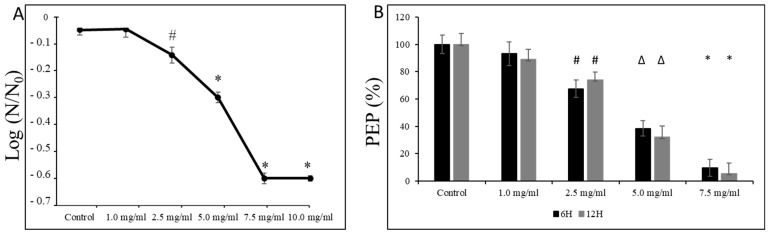
Effect of L-carnitine-fumarate on the aciduricity and PTS of *Streptococcus mutans* UA159. Acid tolerance was determined by measuring the survival of *S. mutans* UA159 at pH = 5.0 assessed on BHI agar plates incubated for 24 h at 37 °C (**A**). PEP level was assessed at mid-logarithmic (6 h) and late logarithmic/early stationary phase (12 h) as described in the Section 2 (**B**). Control—10% 1 × PBS; LC—L-carnitine-fumarate; PEP—phosphoenolpyruvate; # *p* < 0.05, ∆ *p* ≤ 0.01, * *p* < 0.001 compared with control. N_0_ and N—CFU counts before (N_0_, time = 0 h) and after 2 h (N_,_ time = 2 h) treatment in pH = 5.0 culture, respectively.

**Table 1 microorganisms-12-01613-t001:** Antibacterial and antibiofilm effect of L-carnitine forms against oral *Streptococcus* strains.

Test Parameters	L-Carnitine-Fumarate (mg/mL)				
	MIC_90_	MBC_90_	MBIC	MBBC	MBEC
*S. mutans* UA159	6.0	7.0	7.5	10.0	ns
*S. sobrinus* SL1	5.5	6.5	7.5	10.0	ns
*S. mutans* + *S. sobrinus* UA159 + SL1	6.0	7.0	7.5	10.0	ns
	L-carnitine HCl (mg/mL)				
	MIC_90_	MBC_90_	MBIC	MBBC	MBEC
*S. mutans* UA159	6.0	7.0	7.5	10.0	ns
*S. sobrinus* SL1	5.5	6.5	7.5	10.0	ns
*S. mutans* + *S. sobrinus* UA159 + SL1	6.0	7.0	7.5	10.0	ns

ns—not susceptible.

**Table 2 microorganisms-12-01613-t002:** Composition of *S. mutans* UA159 biofilm after treatments with L-carnitine-fumarate.

Test Parameters	Control	2.5 mg/mL	5.0 mg/mL	7.5 mg/mL	10.0 mg/mL
Dry weight (mg/biofilm)	4.41 ± 0.38	4.35 ± 0.31	3.95 ± 0.34	3.62 ± 0.26 *	3.51 ± 0.29 *
Total protein (mg/biofilm)	2.69 ± 0.29	2.48 ± 0.31	2.17 ± 0.28 *	1.93 ± 0.29 *	1.84 ± 0.34 *
Soluble polysaccharides (mg/biofilm)	0.37 ± 0.19	0.36 ± 0.18	0.33 ± 0.04	0.26 ± 0.08 *	0.24 ± 0.03 *
Insoluble polysaccharides (mg/biofilm)	1.19 ± 0.15	1.18 ± 0.16	0.12 ± 0.17	0.89 ± 0.11 *	0.81 ± 0.12 *

* *p* < 0.001 compared to control.

## Data Availability

All data are contained within the manuscript.

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
