# Peer review of "Antibacterial and Antibiofilm Effects of L-Carnitine-Fumarate on Oral Streptococcal Strains Streptococcus mutans and Streptococcus sobrinus"

_microorganisms, 2024, doi:10.3390/microorganisms12081613_

Round 1

Reviewer 1 Report

Comments and Suggestions for Authors

Dear authors,

Thank you for your manuscript which shows that you have done a hard work.

Nevertheless I would like to make a few comments as well I am enclosing a version of the pdf with some corrections.

First of all, S. sobrinus is not a strain of S. mutants but a species in the mutans group of streptococci.

Second, did you check previously if there is any antagonistic or synergistic effect between the 2 streptococci used in your experiments?

Lastly, in general the descriptions of the methods used are quite confusing and sometimes planktonic and biofilm methods are mixed up.

Regards

Author Response

July 31, 2024

Miracle Peng, Section Managing Editor

Microorganisms

Ref: Submission ID - microorganisms-3133029

Title: Antibacterial effect of L-carnitine-fumarate on oral streptococci
strains Streptococcus mutans and Streptococcus sobrinus.

Reviewer 1 comments and concerns:

We would like to thank Reviewer 1 for kind words and identifying specific areas of our manuscript that needed clarifications and adjustments. We highly appreciate it.

“Dear authors, Thank you for your manuscript which shows that you have done a hard work. Nevertheless, I would like to make a few comments as well I am enclosing a version of the pdf with some corrections.”

Q1. First of all, S. sobrinus is not a strain of S. mutans but a species in the mutans group of streptococci.

A1: We corrected this error. Thank you for pointing this out.

Q2. Second, did you check previously if there is any antagonistic or synergistic effect between the 2 streptococci used in your experiments?

A2. We did not do that in this project. However, we noticed that S. sobrinus strain is more susceptible to L-carnitine-fumarate and L-carnitine HCl treatment and that S. mutans strain can “overtake” S. sobrinus (i.e., whenever equilibrium in the two-strain culture is in imbalance, S. mutans tends to expand and overgrow). There are some published reports implying that S. mutans can do that. Nonetheless, we agree with Reviewer 1 that this topic is worth pursuing.

Q3. Lastly, in general the descriptions of the methods used are quite confusing and sometimes planktonic and biofilm methods are mixed up.

A3. We clarified this part in our manuscript. Thank you for pointing this out.

Note: We also addressed (and included in the revision) other issues pointed by Reviewer 1 in provided peer-revied pdf version.

We hope that the revised manuscript is acceptable for further review and acceptance. We look forward to your response.

Sincerely,

Anna Goc, Ph.D.

Senior Research Scientist

Dr. Rath Research Institute

5941 Optical Court, San Jose, CA 95138

Reviewer 2 Report

Comments and Suggestions for Authors

The study covers an important topic regarding the inhibition of the oral caries caused by Streptococcus mutans and another streptococci from mutans group. Oral microbiota is important in the formation of the gastrointestinal microbiota, and is involve in their composition. Due to the growing problem of antibiotic resistance, also among oral bacteria, the searching for new agents with potential antibacterial activity is scientific importance. Noteworthy is the research used in this work, conducted in relation to the most important factor of dental plaque and caries, which is S. mutans, as well as the combination of S. mutans and S. sorbinus. Testing the L-carnitine (in two types) as a potential antibacterial compound is an innovative idea. There is relatively little information on this subject in global literature. Possible works concern on the effects of L-carnitine on the human body. Individual works examined the metabolism of bacteria in relation to L-carnitine and the influence of the L-carnitine on the human gut microbiota.

In my opinion, the work is very interesting and well prepared. The results of the study indicate the potential use of L-carnitine as an antibacterial agent in the prevention of caries, especially in relation to inhibition of the bacterial adhesion and biofilm formation on dental biomaterials.

The manuscript is very well-written, though some minor corrections are needed. The methodology presented in the  study is well chosen and properly explained. The results are presented in detail and accurately. The discussion is comprehensive but could include more current research on L-carnitine. The literature list contains also many current reports. Of course, these are single reports. However, the results of your research compared to those conducted by other authors could significantly increase the validity of the results presented in this publication, especially in terms of the safety of L-carnitine use against the development of caries, and the possibility of using the antibacterial properties of L-carnitine in relation to human oral tissues or only inanimate matter, such as dental biomaterials. I provide a suggestion for additional literature below, for your consideration. I did not find visible stylistic and spelling mistakes, substantive errors or gross linguistic errors in the text.

Minor changes in detail:

  1. The both sections Discussion and References should be added the latest news in the topic.
  2. Please consider adding the current references.

Proposed references:

·       Wu, Qiu et al. “High l-Carnitine Ingestion Impairs Liver Function by Disordering Gut Bacteria Composition in Mice.” Journal of agricultural and food chemistry vol. 68,20 (2020): 5707-5714. doi:10.1021/acs.jafc.9b08313

·       Rajakovich, Lauren J et al. “Elucidation of an anaerobic pathway for metabolism of l-carnitine-derived γ-butyrobetaine to trimethylamine in human gut bacteria.” Proceedings of the National Academy of Sciences of the United States of America vol. 118,32 (2021): e2101498118. doi:10.1073/pnas.2101498118

·       Meadows, Jamie A, and Matthew J Wargo. “Carnitine in bacterial physiology and metabolism.” Microbiology (Reading, England) vol. 161,6 (2015): 1161-74. doi:10.1099/mic.0.000080

·       Koeth, Robert A et al. “l-Carnitine in omnivorous diets induces an atherogenic gut microbial pathway in humans.” The Journal of clinical investigation vol. 129,1 (2019): 373-387. doi:10.1172/JCI94601

Author Response

July 31, 2024

Miracle Peng, Section Managing Editor

Microorganisms

Ref: Submission ID - microorganisms-3133029

Title: Antibacterial effect of L-carnitine-fumarate on oral streptococci
strains Streptococcus mutans and Streptococcus sobrinus.

Reviewer 2 comments and concerns:

We would like to thank Reviewer 2 for identifying the specific areas in our manuscript that need further clarifications and adjustments. We value the opinion of Reviewer 2 very much.

“The study covers an important topic regarding the inhibition of the oral caries caused by Streptococcus mutans and another streptococci from mutans group. Oral microbiota is important in the formation of the gastrointestinal microbiota, and is involve in their composition. Due to the growing problem of antibiotic resistance, also among oral bacteria, the searching for new agents with potential antibacterial activity is scientific importance. Noteworthy is the research used in this work, conducted in relation to the most important factor of dental plaque and caries, which is S. mutans, as well as the combination of S. mutans and S. sobrinus. Testing the L-carnitine (in two types) as a potential antibacterial compound is an innovative idea. There is relatively little information on this subject in global literature. Possible works concern on the effects of L-carnitine on the human body. Individual works examined the metabolism of bacteria in relation to L-carnitine and the influence of the L-carnitine on the human gut microbiota. In my opinion, the work is very interesting and well prepared. The results of the study indicate the potential use of L-carnitine as an antibacterial agent in the prevention of caries, especially in relation to inhibition of the bacterial adhesion and biofilm formation on dental biomaterials. The manuscript is very well-written, though some minor corrections are needed. The methodology presented in the study is well chosen and properly explained. The results are presented in detail and accurately. The discussion is comprehensive but could include more current research on L-carnitine. The literature list contains also many current reports. Of course, these are single reports. However, the results of your research compared to those conducted by other authors could significantly increase the validity of the results presented in this publication, especially in terms of the safety of L-carnitine use against the development of caries, and the possibility of using the antibacterial properties of L-carnitine in relation to human oral tissues or only inanimate matter, such as dental biomaterials. I provide a suggestion for additional literature below, for your consideration. I did not find visible stylistic and spelling mistakes, substantive errors or gross linguistic errors in the text.”

Q1 and 2. Both sections Discussion and References should be added the latest news in the topic. Please consider adding the current references. Proposed references:

  • Wu, Qiu et al. “High l-Carnitine Ingestion Impairs Liver Function by Disordering Gut Bacteria Composition in Mice.” Journal of agricultural and food chemistryvol. 68,20 (2020): 5707-5714. doi:10.1021/acs.jafc.9b08313
  • Rajakovich, Lauren J et al. “Elucidation of an anaerobic pathway for metabolism of l-carnitine-derived γ-butyrobetaine to trimethylamine in human gut bacteria.” Proceedings of the National Academy of Sciences of the United States of Americavol. 118,32 (2021): e2101498118. doi:10.1073/pnas.2101498118
  • Meadows, Jamie A, and Matthew J Wargo. “Carnitine in bacterial physiology and metabolism.” Microbiology (Reading, England)vol. 161,6 (2015): 1161-74. doi:10.1099/mic.0.000080
  • Koeth, Robert A et al. “l-Carnitine in omnivorous diets induces an atherogenic gut microbial pathway in humans.” The Journal of clinical investigationvol. 129,1 (2019): 373-387. doi:10.1172/JCI94601

A1 and 2. We added suggested references with a short paragraph in the Discussion section of our manuscript to accommodate them properly. Thank you for recommending them.

We hope that the revised manuscript is acceptable for further review and acceptance. We look forward to your response.

Sincerely,

Anna Goc, Ph.D.

Senior Research Scientist

Dr. Rath Research Institute

5941 Optical Court, San Jose, CA 95138

Reviewer 3 Report

Comments and Suggestions for Authors

I have gone through the manuscript microorganisms-3133029 - entitled “Antibacterial effect of L-carnitine-fumarate on oral streptococci strains Streptococcus mutans and Streptococcus sobrinus”. Even though the research is of great interest, presentation of results and methods used is very unclear. My biggest concern is that decried methodology is very confusing, with some crucial information missing. Further on, discussion needs to be rewritten, given the fact that in this form it only repeats the results; there is not enough comparison with published literature, or clear explanation of L-carnitine-fumarate mechanisms of action.

I’ve listed here some general comments and suggestions that need to be addressed. Overall, the manuscript needs to go through major changes in order to be considered for publishing.

ABSTRACT

Line 19: Please remove the word mutans from sentence.

Lines 30-33: In this part it was stated what was done, but no results were given. Please change it.

 INTRODUCTION:

Line 39: At the begging of this section please remove from the parentheses S. mutans, no need for it.

Please give more information in the introduction part about why there is a need to test these kind of compounds for the treatment of oral biofilms, aren’t there any commercial agents that are being used?

Also, in Line 80 it is stated that bacteria can utilize carnite, so if they use it in cellular metabolism, how can we use carnite derivates for the inhibition of bacterial growth? Can you explain this part in more details?

At the end of introduction please state what was the main aim of your study.

 MATERIALS AND METHODS

Line 92: Please include the strain label for S. sobrinus.

Clearly state in every described method that you tested effect of L-carnite on both strains separately and then on dual culture. The first time that information is showed is in the results.

The subsections 2.2. and 2.3. can be combined into one, especially cause in the lines 117-126 there is repetition from the 2.2. section. Please combine these sections. Also change biocidal to bactericidal throughout the whole text.

Line 127: Can you explain how is possible to count colonies, cause MBC stands for minimal bactericidal concentration, meaning that 99,9% of bacterial cells are killed.

Please explain why did you do kinetic study of both planktonic and biofilm forms. How does this contribute to research? Also, in the abstract there is no information about these experiments. Furthermore, there is no information about concentration used in subsection 2.4., are the concentrations same as in MIC assay?

Combine subsection 2.5 and 2.6 as suggested for 2.2. and 2.3. subsections.

Can you explain why were concentrations ranging from 2.5-10 mg/ml used in these studies? Were they chosen based on the results of MIC assay?

Lines 173-177: It is stated that viability assay and crystal staining assay were applied. How did you determine the MBBC? It is not possible to detach cells by applying dye, how did you do it?

Lines 178-183: Explain the purpose of recovery assay.

The most confusing parts are subsections 2.8 and 2.9. It is not clear why are these subsections described. In the sections where you determined MIBC and MBBC it is stated that alamarBlue viability assay and crystal violet assay were used. So why describe these methods as separate ones? Please explain.

Subsection 2.11. is also written very confusing. It is titled quantitative determination of EPS,  but at the begging fluorescent imaging of biofilm is described, that is not quantitative, its qualitative method.

Please explain why you did only S. mutans in subsections 2.10., 2.11., 2.12. and 2.13.

 RESULTS

Lines 305-307: It is not clear why L-carnite HCl was used, is it some sort of control? The first mention of this is in the results, it should be in methodology, explained why it was used.

Table1. In the title of Table add antibiofilm, it would be antibacterial and antibiofilm effect of….

Line 399/Figure 4: What do you mean by detachment of biofilms, how was that screened.

 DISCUSSION

As I stated before, the discussion must be expanded.

Line 484: add e to Th

Line 490: add S to L1

Lines 523-525: This sentence cannot be a part of conclusion, is not something that you showed in this study. Focus on your results and draw the appropriate conclusion.

Author Response

July 31, 2024

Miracle Peng, Section Managing Editor

Microorganisms

Ref: Submission ID - microorganisms-3133029

Title: Antibacterial effect of L-carnitine-fumarate on oral streptococci
strains Streptococcus mutans and Streptococcus sobrinus.

Reviewer 3 comments and concerns:

We would like to thank Reviewer 3 for identifying the specific areas in our manuscript that need further clarifications and adjustments.

“I have gone through the manuscript microorganisms-3133029 - entitled “Antibacterial effect of L-carnitine-fumarate on oral streptococci strains Streptococcus mutans and Streptococcus sobrinus”. Even though the research is of great interest, presentation of results and methods used is very unclear. My biggest concern is that decried methodology is very confusing, with some crucial information missing. Further on, discussion needs to be rewritten, given the fact that in this form it only repeats the results; there is not enough comparison with published literature, or clear explanation of L-carnitine-fumarate mechanisms of action. I’ve listed here some general comments and suggestions that need to be addressed. Overall, the manuscript needs to go through major changes in order to be considered for publishing.”

Q1. ABSTRACT

Q1a. Line 19: Please remove the word mutans from sentence.

A1a. S. sobrinus is a species that belongs to the mutans group of streptococci – this was pointed out by Reviewer 1 and is also widely and well-acknowledged fact by other research groups. We believe that this is very important and correct information, a foundation for why we chose these two species and why we examined them. We exchanged the word strain into species to correct the error.

Q1b. Lines 30-33: In this part it was stated what was done, but no results were given. Please change it.

A1b. We corrected this error. Thank you.

Q2. INTRODUCTION

Q2a. Line 39: At the begging of this section please remove from the parentheses S. mutans, no need for it.

A2a. We comply with this request and remove it.

Q2b. Please give more information in the introduction part about why there is a need to test these kind of compounds for the treatment of oral biofilms, aren’t there any commercial agents that are being used?

A2b. We added a short description to the Introduction section of our manuscript to address Reviewer’s 3 request. We are not aware of any commercial agents containing L-carnitine derivatives that are being used publicly yet, but we are not excluding the possibility that they are in the process of introducing them to the market.

Q2c. Also, in Line 80 it is stated that bacteria can utilize carnitine, so if they use it in cellular metabolism, how can we use carnitine derivates for the inhibition of bacterial growth? Can you explain this part in more details?

A2c. We agree with Reviewer 3 that this part is very interesting and warrants further study. Yes, L-carnitine is utilized by many bacterial strains, but not necessarily its derivatives. Also, concentration matters. It looks like lower concentrations of L-carnitine forms/derivatives have no bacteriostatic or bactericidal effect, but in contrast, do have on other processes relevant to bacterial patho-physiology. We used two different forms of L-carnitine to check whether obtained results are form-specific or not. We also considered the fact that we wanted to use compounds that are not virtuously synthetic. We are not aware of studies that report about applicability of different forms of L-carnitine or its derivatives. There is not so much data about antibacterial effects of them yet. We cannot exclude that in near future such compounds/derivatives will find their way to the public though. We also did not explore the mechanism of action tested derivatives, but presented results from the experiment that may serve as a clue for further studies focusing on more details addressing this issue. Finally, formulation of L-carnitine derivatives seems to matter to make them “bench to bed site” agents/treatment(s). This topic (anti-bacterial and antibiofilm properties of L-carnitine forms/derivatives) still requires more research that would substantiate any claims and avoid any speculations. 

Q2d. At the end of introduction please state what was the main aim of your study.

A2d. We added the requested information to the Introduction section of our manuscript. Thank you for this suggestion.

Q3. MATERIALS AND METHODS

Q3a. Line 92: Please include the strain label for S. sobrinus.

A3a. We added this requested information.

Q3b. Clearly state in every described method that you tested effect of L-carnitine on both strains separately and then on dual culture. The first time that information is showed is in the results.

A3b. We clarified this part of our manuscript per Reviewer’s 3 request.

Q3c. The subsections 2.2. and 2.3. can be combined into one, especially cause in the lines 117-126 there is repetition from the 2.2. section. Please combine these sections. Also change biocidal to bactericidal throughout the whole text.

A3c. We comply with these requests.

Q3d. Line 127: Can you explain how is possible to count colonies, cause MBC stands for minimal bactericidal concentration, meaning that 99,9% of bacterial cells are killed.

A3d. We used methodology that is explained in the details in publication by Wiegand, I.; Hilpert, K.; Hancock, R.E.W. Agar and broth dilution methods to determine the minimal inhibitory concentration (MIC) of antimicrobial substances. Nat Protocols. 2008, 3, 163–175. In short, serial dilutions of culture are applied on agar (in our case BHI agar) plates and incubated for 24h in order to count (with colony counter instrument) and calculated according to algorithm acknowledged as a standard in th microbiology field.

Q3e. Please explain why did you do kinetic study of both planktonic and biofilm forms. How does this contribute to research? Also, in the abstract there is no information about these experiments. Furthermore, there is no information about concentration used in subsection 2.4., are the concentrations same as in MIC assay?

A3e. Kinetic study helped us to establish timing necessary to perform other experiments presented in this study, but also provided more information on how fast applied L-carnitine forms work against tested strains. Since Abstract has a limited space and number of characters, we focused only on providing consolidated information crucial to communicate the main findings. Information about used concentrations (as stated in the manuscript) is provided in the Figure legend.

Q3f. Combine subsection 2.5 and 2.6 as suggested for 2.2. and 2.3. subsections.

A3f. We comply with this request.

Q3g. Can you explain why were concentrations ranging from 2.5-10 mg/ml used in these studies? Were they chosen based on the results of MIC assay?

A3g. Yes, they were chosen based on the results of MIC assay as well as initial preliminary screening testing.

Q3h. Lines 173-177: It is stated that viability assay and crystal staining assay were applied. How did you determine the MBBC? It is not possible to detach cells by applying dye, how did you do it?

A3h. We corrected this error. Thank you very much for pointing this out.

Q3i. Lines 178-183: Explain the purpose of recovery assay.

A3i. We used alamarBlue day to check the viability of the biofilm. We chose this method precisely to avoid any detachment of biofilm from the plate. Although it is practiced and done by scratching or sonication followed by plating detached biofilm on agar plates and further incubations for any regrow/recovery in the form of colonies, we were concerned that this can somehow affect viability of the biofilm per se. Thus, we decided to use a less invasive method by directly applying alamarBlue (ready-to-use day used by us and other research groups for this purpose) to check viability status/metabolic rate. For this reason, we first treated biofilm with LC, next removed the old medium containing BHI broth and LC and replaced it with BHI broth only, leaving it in the incubator for further 24h in order to check if any regrow/recovery from applied earlier treatment will happen. That was the purpose of the recovery assay.

Q3j. The most confusing parts are subsections 2.8 and 2.9. It is not clear why are these subsections described. In the sections where you determined MIBC and MBBC it is stated that alamarBlue viability assay and crystal violet assay were used. So why describe these methods as separate ones? Please explain.

A3j. We used alamarBlue as a dye to validate viability of biofilm. AlamarBlue reagent is a resazurin-based dye for cytotoxicity studies (i.e., based on status of metabolic activity; Assess the Cell Viability of Staphylococcus aureus Biofilms | Thermo Fisher Scientific - US). And was used not only by us but also other research groups for this particular purpose (e.g., Pettit et al. Application of a high throughput Alamar blue biofilm susceptibility assay to Staphylococcus aureus biofilms. Ann. Clin. Microbiol. Antimicrob. 2009). Crystal violet, although is used to test viability, alaramBlue gives more reproducible and reliable results. Nonetheless, crystal violet is very useful to check amount of attached biofilm, and acknowledged for that, since it corresponds very well with what is seen on the plate (e.g., Kamimura et al. Quantitative Analyses of Biofilm by Using Crystal Violet Staining and Optical Reflection. Materials. 2022; Wilson et al. Quantitative and Qualitative Assessment Methods for Biofilm Growth: A Mini-review. Res. Rev. J. Eng. Technol. 2017).

Q3k. Subsection 2.11. is also written very confusing. It is titled quantitative determination of EPS, but at the begging fluorescent imaging of biofilm is described, that is not quantitative, its qualitative method.

A3k. We correct this error. Thank you very much for pointing this out.

Q3l. Please explain why you did only S. mutans in subsections 2.10., 2.11., 2.12. and 2.13.

A3l. We used S. mutans UA159 strain in subsections 2.10., 2.11., 2.12. and 2.13 as an example of the most relevant strain in dental plaque and as a model strain used to examine antibacterial effects and also to examine the effect of tested L-carnitine forms on processes important to its virulence and prevalence in dental caries and plaque.

Q4. RESULTS

Q4a. Lines 305-307: It is not clear why L-carnitine HCl was used, is it some sort of control? The first mention of this is in the results, it should be in methodology, explained why it was used.

A4a. We used two different forms of L-carnitine to check whether the effect we observed is form-specific or not. Also, L-carnitine-fumarate is more stable in the culture and less hydrophilic than L-carnitine HCl, but L-carnitine HCl is better to dissolve, may have less safety issues and is less modified version of L-carnitine itself. We comply with Reviewer 3 request and added information about L-carnitine HCl to the Methodology section of our manuscript. Thank you for pointing this out.

Q4b. Table1. In the title of Table add antibiofilm, it would be antibacterial and antibiofilm effect of….

A4b. We corrected this error. Thank you for pointing this out.

Q4c. Line 399/Figure 4: What do you mean by detachment of biofilms, how was that screened.

A4c. We used crystal violet to check how much biofilm was left after applied L-carnitine treatment. We and other research groups as well noticed that the amounts of the extracted day correspond well with the amount of biofilm left on the plate.

Q5. DISCUSSION

Q5a. As I stated before, the discussion must be expanded.

A5a. We comply with this request to the best of our ability.

Q5b. Line 484: add e to Th

A5b. We corrected this error.

Q5c. Line 490: add S to L1

A5c. We corrected this error.

Q5d. Lines 523-525: This sentence cannot be a part of conclusion, is not something that you showed in this study. Focus on your results and draw the appropriate conclusion.

A5d. We complied with this request and removed it from the Conclusion section of our manuscript.

We hope that the revised manuscript is acceptable for further review and acceptance. We look forward to your response.

Sincerely,

Anna Goc, Ph.D.

Senior Research Scientist

Dr. Rath Research Institute

5941 Optical Court, San Jose, CA 95138

Round 2

Reviewer 1 Report

Comments and Suggestions for Authors

Dear authors thanks for correcting the manuscript.

Reviewer 3 Report

Comments and Suggestions for Authors

The authors responded clearly to all suggestions. In this form, I would recommend manuscript for publication.